

# Phylogeography of *Arabidopsis halleri* (Brassicaceae) in mountain regions of Central Europe inferred from cpDNA variation and ecological niche modelling

Pawel Wasowicz[1,2], Maxime Pauwels[3], Andrzej Pasierbinski[2], Ewa M. Przedpelska-Wasowicz[4], Alicja A. Babst-Kostecka[5], Pierre Saumitou-Laprade[6] and Adam Rostanski[2]

[1] Icelandic Institute of Natural History, Iceland

[2] Faculty of Biology and Environmental Protection, Department of Botany and Nature Protection, University of Silesia, Katowice, Poland

[3] Unité Evo-Eco-Paléo (EEP)—UMR 8198, Université de Lille—Sciences et Technologies, CNRS, Villeneuve d'Ascq, France

[4] Institute of Botany, University of Warsaw, Warszawa, Poland

[5] Department of Ecology, Institute of Botany, Polish Academy of Sciences, Krakow, Poland

[6] Unité Evo-Eco-Paléo (EEP)—UMR 8198, Université des Sciences et Technologies de Lille (Lille I), Villeneuve d'Ascq, France

Corresponding author
Pawel Wasowicz,
pawasowicz@gmail.com

## ABSTRACT

The present study aimed to investigate phylogeographical patterns present within *A. halleri* in Central Europe. 1,281 accessions sampled from 52 populations within the investigated area were used in the study of genetic variation based on chloroplast DNA. Over 500 high-quality species occurrence records were used in ecological niche modelling experiments. We evidenced the presence of a clear phylogeographic structure within *A. halleri* in Central Europe. Our results showed that two genetically different groups of populations are present in western and eastern part of the Carpathians. The hypothesis of the existence of a glacial refugium in the Western Carpathians adn the Bohemian Forest cannot be rejected from our data. It seems, however, that the evidence collected during the present study is not conclusive. The area of Sudetes was colonised after LGM probably by migrants from the Bohemian Forest.

## INTRODUCTION

A phylogeographical approach has been used in numerous studies addressing the Quaternary history of the flora of Europe, shaped by repeated range contractions during cold periods and subsequent extension of available habitats during warmer periods (*Hewitt, 2000*; *Hewitt, 2004*). These range oscillations, altering the patterns of gene flow, have been found to contribute to the genetic differentiation that can be detected between

contemporary populations (*Médail & Diadema, 2009*). The presence of this differentiation has allowed (with the advent of phylogeography) insights into processes responsible for range formation, reconstruction of (re)colonisation routes, detection of refugial areas and unravelling historical relationships among different parts of the contemporary species distribution.

Areas of relative ecological stability that provided suitable habitats for species survival during periods of glaciation are termed refugia (*Tribsch & Schönswetter, 2003*). Numerous studies carried out so far showed that major glacial refugia were located in the southern part of Europe (*Taberlet et al., 1998*; *Hewitt, 1999*). Recently, the possibility of full-glacial survival of temperate species at northern latitudes in so-called northern or cryptic refugia (*Stewart & Lister, 2001*) was hypothesised and subsequently supported by fossil records from several species (*Willis & Van Andel, 2004*). There is, however, still little molecular evidence for the existence of "northern refugia" in Central Europe (*Daneck et al., 2011*).

In Europe the potential for phylogeographical research has been exploited intensively in the Alps, where the abundance and complexity of the available phylogeographic studies has already resulted in synthetic and comparative analyses (*Schönswetter et al., 2005*; *Alvarez et al., 2009*). The situation is quite different in other mountain ranges of Central Europe such as the Carpathians, Sudetes, Bohemian Forest (Sumava) and Harz Mts. A recent literature review pointed out clearly that the history of species range formation in the mountains of Central Europe is still only poorly known from phylogeographical studies (*Ronikier, 2011*).

*Arabidopsis halleri* with its pattern of occurrence covering nearly all mountain regions of Central Europe (*Jalas & Suominen, 1994*) seems to be an interesting model taxon to address all the problems raised above. Previous phylogeographical studies focused on the species, evidenced the presence of two major units within the species range and attributed the emergence of these units to vicariance associated with the isolation of two large populations groups during the Quaternary, which were located in Central and Southern Europe (*Pauwels et al., 2012*). The sampling density adopted in the study was, however, too low to address the question of *A. halleri* range formation in the mountains of Central Europe.

In the light of these considerations, we focused our study on the poorly investigated area covering the Carpathians, Sudetes, Bohemian Forest and Harz Mountains in order to reconstruct the phylogeographic history of this montane species *Arabidopsis halleri*. This approach allowed us to overcome another limitation originating from the fact that the vast majority of phylogeographical analyses carried out hitherto on mountain species in Europe has been focused on alpine and arctic-alpine species, while our knowledge of the phylogeographic patterns present within herbaceous species having the centre of its occurrence in lower vegetation belts (subalpine and montane) remains, with some notable exceptions (*Despres, Loriot & Gaudeul, 2002*; *Stachurska-Swakoń, Cieślak & Ronikier, 2013*), much poorer.

We aimed to answer the following questions:

1. Is there any geographically structured genetic variation in *A. halleri* within the investigated area?

2. Is there any evidence for the existence of glacial refugia in Central Europe?
3. What is the origin of *A. halleri* populations in the Carpathians and Sudetes?

## MATERIALS & METHODS

### The study species

*Arabidopsis halleri* (L.) O'Kane & Al-Shehbaz is a perennial, self-incompatible and highly outcrossing (*Llaurens et al., 2008*), stoloniferous herb with a highly disjunctive distribution between Europe and the Far East. It occurs in mountain and upland environments on slopes, forest margins, rocky crevices and river banks from 200 to 2,200 m a.s.l. In Europe, it is widely distributed in the Alps, Carpathians, Sudetes and Dinaric Alps (*Jalas & Suominen, 1994*). Its distribution also covers some upland regions north from the Alps (including the Harz Mountains) and the Western Carpathians (*Jalas & Suominen, 1994*). The species is highly variable in leaf morphology, flower colour and traits connected with the development of stolons. At least three subspecies are quite distinct in terms of morphological variation (*Al-Shehbaz & O'Kane, 2002*): *A. halleri* subsp. *halleri* and *A. halleri* subsp. *ovirensis* occur in Europe, whereas *A. halleri* subsp. *gemmifera* occurs in the Far East. According to other authors (*Kolník & Marhold, 2006*) the species can be alternatively divided into five distinct morphological subspecies, with four occurring in Europe. Only diploid (2n = 2X = 16) individuals have so far been reported from throughout the distribution range (*Al-Shehbaz & O'Kane, 2002*; *Kolník & Marhold, 2006*). Other species from the genus *Arabidopsis* also co-occur within the investigated area (*A.thaliana*, *A.arenosa*, *A.neglecta*, *A. lyrata*). A recent study has shown that reticulation and hybridization among lineages that might have transferred cpDNA types from one lineage into the other is unlikely (*Koch & Matschinger, 2007*). The same authors have pointed out that cpDNA type diversity may predate separation of the main evolutionary lineages within the genus (*Koch & Matschinger, 2007*).

### Sampling and DNA extraction

We sampled 1,281 individuals from 52 populations (Table 1) scattered across the species range in seven geographic regions of Central and Eastern Europe (see Fig. S1). Twenty-five populations included in the present study have been already described in previous studies (*Pauwels et al., 2005*; *Pauwels et al., 2012*). In the present study, sampling was extended to improve sampling representativeness in the area of the Sudetes as well as the Carpathians. All the locations sampled belonged to the native species range (*Jalas & Suominen, 1994*). In each locality, individual samples were taken from plants separated by at least three metres to avoid sampling clones (*Van Rossum et al., 2004*). Sample size generally reflected population size and was almost exhaustive in small populations. Permits for plant sampling were obtained from the following national parks: Krkonossky narodni park (CZ, permit no. 01004/2009), Karkonoski Park Narodowy (PL permits no. 24/2007, 2/2008)/, Tatrzanski Park Narodowy (PL, permit no. NB-056/119/07), Bieszczadzki Park Narodowy (PL, permits no. 54/06, 40/07) and Karpatskij biosfernij Zapovidnik (UA, permit no. 01110607/07). In other cases sampling was done outside protected areas.

**Table 1** Location of sampled populations and sample sizes.

| Population | Locality; collectors | GPS coordinates | | Altitude | |
| | | Latitude | Longitude | (m a.s.l.) | *n* |
| --- | --- | --- | --- | --- | --- |
| A05 | Mutters, Alps, Northern Tyrol, Austria; MP, PSL | 47°13′46.68″ | 11°22′46.72″ | 807 | 14 |
| A08 | W from Mehrn, Alps, Northern Tyrol, Austria; MP, PSL | 47°25′10.56″ | 11°51′57.46″ | 522 | 45 |
| A09 | W from Mehrn, Alps, Northern Tyrol, Austria; MP, PSL | 47°25′14.57″ | 11°51′53.71″ | 519 | 20 |
| CZ04 | SW from Vimperk, Bohemian Forest, Czech Rep.; MP, PSL | 49°02′08.72″ | 13°45′08.41″ | 772 | 14 |
| CZ05 | N from Kubova Huť, Bohemian Forest, Czech Rep.; MP, PSL | 48°59′15.54″ | 13°46′23.40″ | 998 | 24 |
| CZ06 | Kubova Huť, Bohemian Forest, Czech Rep.; MP, PSL | 48°59′00.00″ | 13°46′00.00″ | 1,060 | 12 |
| CZ14 | near Starý Herštajn, Bohemian Forest, Czech Rep.; MP, PSL | 49°28′37.19″ | 12°42′92.15″ | 842 | 20 |
| CZ16 | Horská Kvilda, Bohemian Forest, Czech Rep.; MP, PSL | 49°03′21.05″ | 13°33′18.19″ | 1,052 | 57 |
| CZ18 | NW from Zhuři, Bohemian Forest, Czech Rep.; MP, PSL | 49°05′42.14″ | 13°32′10.31″ | 1,039 | 9 |
| CZ20 | Labská, Sudetes, Czech Rep.; PW, EPW | 50°42′55.9″ | 15°35′00.9″ | 698 | 33 |
| CZ21 | Herlikovice, Sudetes, Czech Rep.; PW, EPW | 50°39′41.6″ | 15°35′44.5″ | 555 | 30 |
| CZ22 | Rýchorská Bouda, Sudetes, Czech Rep.; PW, EPW | 50°39′29.4″ | 15°51′00,00″ | 995 | 32 |
| D01 | NE from Ramspau, Bohemian Forest, Germany; MP, PSL | 49°10′06.40″ | 12°09′08.80″ | 345 | 7 |
| D02 | near Hirschling, Bohemian Forest, Germany; MP, PSL | 49°11′31.00″ | 12°09′52.00″ | 452 | 8 |
| D03 | W from Cham, Bohemian Forest, Germany; MP, PSL | 49°13′10.00″ | 12°39′66.00″ | 362 | 9 |
| D04 | S from Hochfeld, Bohemian Forest, Germany; MP, PSL | 49°09′50.95″ | 12°47′45.43″ | 383 | 11 |
| D08 | S from Oker, Harz Mts., Germany; MP, PSL | 51°53′47.45″ | 10°29′23.97″ | 279 | 12 |
| D09 | S from Glosar, Harz Mts., Germany; MP, PSL | 51°53′27.55″ | 10°25′05.62″ | 325 | 18 |
| D11 | E from Hahnemklee, Harz Mts., Germany; MP, PSL | 51°51′16.17″ | 10°21′56.68″ | 644 | 18 |
| D12 | SE from Lautenthal, Harz Mts., Germany; MP, PSL | 51°51′54.09″ | 10°17′53.85″ | 415 | 18 |
| D13 | SW from Langelsheim, Harz Mts., Germany; MP, PSL | 51°55′13.40″ | 10°18′29.68″ | 231 | 20 |
| D14 | SE from Heersum, Harz Mts., Germany; MP, PSL | 52°06′08.66″ | 10°06′57.62″ | 89 | 11 |
| PL02 | Żyglinek, Western Carpathian Foreland, Poland; MP, PSL | 50°29′40.81″ | 18°56′40.29″ | 298 | 21 |
| PL03 | W from Żyglinek, Western Carpathian Foreland, Poland; MP, PSL | 50°29′30.34″ | 18°57′34.57″ | 302 | 15 |
| PL07 | Ujków Stary, Western Carpathian Foreland, MP, PSL | 50°17′00.92″ | 19°29′03.10″ | 325 | 19 |
| PL08 | N from Chobot, Western Carpathian Foreland, Poland; MP, PSL | 50°05′51.87″ | 20°22′32.89″ | 193 | 12 |
| PL32 | Kościelisko, Western Carpathians, Poland; AK | 49°16′27.72″ | 19°52′45.58″ | 990 | 27 |
| PL33 | Zakopane, Western Carpathians, Poland; AK | 49°17′34.02″ | 19°55′34.59″ | 879 | 21 |
| PL37 | Szklarska Poręba, Sudetes, Poland; PW, EPW | 50°49′08.50″ | 15°31′24.08″ | 690 | 40 |
| PL38 | Orle, Sudetes, Poland; PW, EPW | 50°49′00.59″ | 15°22′51.85″ | 828 | 29 |
| PL39 | E from Kowary, Sudetes, Poland; PW, EPW | 50°47′23.66″ | 15°51′55.82″ | 583 | 39 |
| PL40 | Kowarska Pass, Sudetes, Poland; PW, EPW | 50°45′37.77″ | 15°52′04.37″ | 729 | 39 |
| PL41 | Hala Izerska, Sudetes, Poland; PW, EPW | 50°50′52.36″ | 15°21′45.62″ | 837 | 39 |
| PL42 | Łabski Szczyt, Sudetes, Poland; PW, EPW | 50°47′14.47″ | 15°32′17.18″ | 1,189 | 39 |
| PL43 | Mały Staw, Sudetes, Poland; PW, EPW | 50°44′54.90″ | 15°42′09.56″ | 1,199 | 39 |
| PL44 | W from Zieleniec, Sudetes, Poland; PW, EPW | 50°22′54.28″ | 16°21′47.26″ | 755 | 41 |
| PL45 | Zawadzkie, Western Carpathian Foreland, Poland; PW, EPW | 50°37′23.06″ | 18°26′39.06″ | 208 | 39 |
| PL46 | Sianki, Eastern Carpathians, Poland; PW, EPW | 49°01′14.40″ | 22°53′08.40″ | 800 | 23 |
| PL47 | Roztoki Górne, Eastern Carpathians, Poland; PW, EPW | 49°09′04.30″ | 22°19′16.90″ | 728 | 24 |
| PL48 | Krzywe, Eastern Carpathians, Poland; PW, EPW | 49°11′64.00″ | 22°21′28.80″ | 647 | 20 |

**Table 1** (*continued*)

| Population | Locality; collectors | GPS coordinates | | Altitude | |
| | | Latitude | Longitude | (m a.s.l.) | *n* |
|---|---|---|---|---|---|
| PL49 | Nasiczne, Eastern Carpathians, Poland; PW, EPW | 49°10′22.90″ | 22°35′50.80″ | 643 | 22 |
| PL50 | Łężyce, Sudetes, Poland; PW, EPW | 50°26′44.8″ | 16°20′58.3″ | 712 | 34 |
| SK02 | Vysný Klátov, Western Carpathians, Slovakia; MP, PSL | 48°46′10.33″ | 21°07′48.49″ | 586 | 22 |
| SK05 | NE from Javorina, Western Carpathians, Slovakia; MP, PSL | 49°16′59.11″ | 20°09′14.51″ | 994 | 46 |
| SK10 | Štrbské Pleso, Western Carpathians, Slovakia; PW, EPW | 49°07′07.26″ | 20°03′42.24″ | 1,322 | 21 |
| SK12 | Plihov, Western Carpathians, Slovakia; PW, EPW | 49°24′15.30″ | 20°42′08.52″ | 466 | 16 |
| UA01 | Rakhiv, Eastern Carpathians, Ukraine; PW, AR, PSL | 48°01′31.60″ | 24°10′02.80″ | 434 | 31 |
| UA02 | Kvasy, Eastern Carpathians, Ukraine; PW, AR, PSL | 48°07′59.00″ | 24°16′26.10″ | 530 | 32 |
| UA04 | Vil'shany, Eastern Carpathians, Ukraine; PW, AR, PSL | 48°19′57.60″ | 23°36′22.80″ | 555 | 32 |
| UA05 | Synevyr, Eastern Carpathians, Ukraine; PW, AR, PSL | 48°32′04.80″ | 23°38′53.50″ | 699 | 32 |
| UA06 | Synevyr Lake, Eastern Carpathians, Ukraine; PW, AR, PSL | 48°37′01.20″ | 23°41′10.20″ | 1,006 | 16 |
| UA07 | Synevyrska Poliana, Eastern Carpathians, Ukraine; PW, AR, PSL | 48°36′00.60″ | 23°41′54.30″ | 829 | 8 |

**Notes.**

*n*, sample size.

Collected leaf tissue was immediately dried in silica gel prior to molecular analysis. DNA was extracted from 10 to 15 mg of dry material using NucleoSpin® 96 Plant (Macherey-Nagel), and PCR amplifications were performed on 1:20 dilutions.

## Genetic analysis
### Genotyping procedure
During the genotyping process we screened 12 polymorphic sites previously detected by PCR-RFLP in three cpDNA regions: *trn*K intron and two intergenic regions (*trn*C-*trn*D, *psb*C-*trn*S). The observed polymorphisms are briefly characterized in Table 2. In contrast to *Pauwels et al. (2005)*, however, three different genotyping methods were employed.

### SNaPshot assay
After PCR amplification of the three cpDNA regions of interest, we used SNaPshot assay for simultaneous detection of seven single nucleotide polymorphisms (SNP, cf. Table 2). In the analysis we employed ABI Prism® SNaPshot® Multiplex Kit (Applied Biosystems) and followed the protocol given by the manufacturer.

The following PCR conditions were utilized: a total volume of 15 µl consisting of 3 µl of template DNA (20–100 ng), 6.225 µl of water, 1.5 µl of 10X PCR Buffer, 2.1 µl of 25 mM solution of $MgCl_2$, 1.2 µl of 2.5 mM solution of dNTP, 0.3 µl of BSA solution (10 mg/ml), 0.3 µl of 10 mM solution of each primer and 0.075 µl of AmpliTaq®DNA Polymerase (5U/µl). The reaction was carried out in Mastercycler® ep gradient S thermal cycler using one cycle of 5 min at 95 °C, and 36 cycles of 45 s at 92 °C, 45 s at 58 °C–62 °C (depending of the primers sequences, precise protocol upon request); and 2 min 30 s at 72 °C, followed by one cycle of 10 min at 72 °C. The primer sequences for specific PCR amplifications are given in Table S1. Amplicons were used for genotyping ollowing the manufacturer's instruction (ABI Prism® SNaPshot® Multiplex Kit). The primer sequences used in the SNaPshot reaction are given in Table S2. Sample electrophoresis was carried out in an

Wasowicz et al. (2016), *PeerJ*, DOI 10.7717/peerj.1645

**Table 2  cpDNA polymorphism observed in investigated material.** Observed mutations were numbered from 1 to 12 (numbers correspond with those given in Fig. 1 and in Table 3).

| Mutation number | RFLP (PAUWELS I IN. 2005) | Mutation type | Methodology | Allelic variation | | Coding | Fragment size (bp) |
|---|---|---|---|---|---|---|---|
| | | | | Base detected (SNaPshot) | Polymorphism type | | |
| 1 | K1K2 HpaII3 | SNP | SNaPshot | A | *ATC*A*GG* | 1 | – |
| | | | | C | *ATC*T*GG* | 9 | – |
| 2 | K1K2 AscI3 | SNP | SNaPshot | T | *AGG*T*AT* | 1 | – |
| | | | | A | *AGG*A*AT* | 9 | – |
| 3 | K1K2 Tru9I5 | SNP | SNaPshot | G | *TT*G*AAT* | 1 | – |
| | | | | T | *TT*T*AAT* | 9 | – |
| 4 | CS AluI1 | SNP | SNaPshot | C | *XTAGC*C*ACTT* | 1 | – |
| | | | | T | *XTAGC*T*ACTT* | 9 | – |
| 5 | CS HinfI4 | SNP | SNaPshot | C | *XAATA*C*ACTC* | 1 | – |
| | | | | G | *XAATA*G*ACTC* | 9 | – |
| 6 | CD AscIA | SNP | CAPS | – | *X*T*AATTT* | 1 | 908 |
| | | | | – | *X*A*AATTT* | 9 | 731 + 171 |
| 7 | CD AscIB | SNP | CAPS | – | *XGAA*A*TTN* | 1 | 908 |
| | | | | – | *XGAA*T*TT* | 9 | 613 + 295 |
| 8 | CD AscI4 | SNP | CAPS | – | *XGAA*A*TT* | 1 | 497 |
| | | | | – | *XGAA*T*TT* | 9 | 351 + 146 |
| 9 | K1K2HinfI6B | SNP | SNaPshot | T | *TTT*T*AT* | 1 | – |
| | | | | G | *TTT*G*AT* | 9 | – |
| 10 | K1K2 HinfI5A | INDEL | SNaPshot | T | **ATATCT***TATTCTTATTG* | 1 | – |
| | | | | A | **A**——— *TATTCTTATTC* | 2 | – |
| 11 | K1K2 Tru9IB | INDEL | length poly-morphism | – | *X*AAA**TAACT***TTTTTGT* | 1 | 227 |
| | | | | – | *X*AAA——— *TTTTTGT* | 2 | 222 |
| 12 | CD HinfI8A | INDEL | Length poly-morphism | – | *X*GGA**TTTTTTTTTTAGA***AAT* | 1 | 81 |
| | | | | – | *X*GGA—————————AAT | 2 | 69 |

ABI PRISM® 3,130 xl Genetic Analyzer (Applied Biosystems) using a capillary of 36 cm, POP 4 Migration Buffer and Dye Set E5. The data were collected using Foundation Data Collection 4.0 software and analyzed with GeneMapper 4.0.

### CAPS assay

The CAPS assay was used for genotyping three SNP polymorphisms in the *trn*C-*trn*D region by specifically amplifying the genomic region containing the SNP polymorphisms mentioned in Table 2 and digesting the amplification product using the *Acs*I restriction enzyme. The PCR primer sequences were given in Table S2. The PCR mixture and the conditions followed the protocol given above for SNaPshot assay. Restriction enzyme reaction was performed on a total volume of 20 µl consisting of 10 µl of PCR product and 10 µl of restriction mixture containing 5.9 µl of water, 2 µl of SuRe/Cut Buffer B 10X, 2 µl of 2 mM solution of spermidine and 0.1 µl of AcsI solution (10U/µl). The mixture was then incubated at 50 °C for 1 h, followed by enzyme deactivation at 70 °C for 15 min. Both procedures were carried out in Mastercycler® ep gradient S (Eppendorf) thermal cyclers. Fragments were separated by electrophoresis on 2% agarose gels stained by ethidium bromide and photographed with BioImage system (Bioprobe) under UV light.

### PCR length difference assay

Two indel polymorphisms in the *trn*K and *trn*C-*trn*D regions (cf. Table 2) were genotyped from a PCR product length difference assay following the method described by *Oetting et al. (1995)*. The PCR mixture followed the protocol given above for SNaPshot assay. The primer sequences used in this reaction are given in Table S1. PCRs were carried out in Mastercycler® ep gradient S (Eppendorf) thermal cyclers using one cycle of 5 min at 95 °C, and 36 cycles of 30 s at 94 °C, 30 s at 51 °C or 56 °C (depending of the primers sequences); and 30 s at 72 °C, followed by one cycle of 10 min at 72 °C. Fragments were separated on 8% polyacrylamide gels using a Li-Cor 4200 Global IR2 DNA Sequencer.

## Data analysis

Since cpDNA is a non-recombining molecule, alleles observed at all twelve loci were combined into cpDNA haplotypes (chlorotypes, Table 3). Chlorotype nomenclature is fully consistent with our previous study (*Pauwels et al., 2005*).

### Phylogenetic relationships between chlorotypes

A minimum spanning tree (MST) was constructed on the basis of a distance matrix reflecting molecular differences between each pair of chlorotypes using a modification of the algorithm described by *Rohlf (1973)*. Computations were made using the software Arlequin 3.11 (*Excoffier, Laval & Schneider, 2005*). The MST algorithm assumes that each chlorotype is linked to all the other chlorotypes by one or a series of mutations and constructs a tree with minimal number of required mutational steps between haplotypes (*Excoffier & Smouse, 1994*). The position of a chlorotype in the MST also gives information regarding relative age, since older haplotypes are expected to locate in internal nodes of the tree (*Posada & Crandall, 2001*).

Wasowicz et al. (2016), *PeerJ*, DOI 10.7717/peerj.1645

**Table 3 Description of cpDNA chlorotypes identified in investigated populations of *A. halleri*.** Characters used in coding correspond to coding column in Table 2. Correspondence to the mutations observed by *Pauwels et al. (2005)* in RFLP study was given. Mutation numbers corresponds with Fig. 1.

| | Mutation number | 1 | 2 | 3 | 4 | 5 | 6 | 7 | 8 | 9 | 10 | 11 | 12 |
|---|---|---|---|---|---|---|---|---|---|---|---|---|---|
| | Mutation name (cf. *Pauwels et al., 2005*) | K1K1HpaII3 | K1K2 AcsI3 | K1K1 Tru9I5 | CS Alu I1 | CS HinfI4 | CD, AcsI 1A | CD AcsIB | CD AcsI4 | K1K2 HinfI6B | K1K2, HinfI5A | K1K2 Tru9IB | CD HinfI8A |
| Chlorotype | A | 9 | 9 | 9 | 1 | 1 | 1 | 1 | 9 | 1 | 1 | 1 | 1 |
| | B | 9 | 9 | 9 | 9 | 1 | 1 | 1 | 9 | 1 | 1 | 1 | 1 |
| | C | 9 | 9 | 9 | 1 | 1 | 1 | 9 | 9 | 1 | 1 | 1 | 1 |
| | D | 9 | 1 | 9 | 1 | 1 | 1 | 1 | 9 | 1 | 1 | 1 | 1 |
| | E | 1 | 9 | 9 | 1 | 1 | 1 | 1 | 9 | 1 | 1 | 1 | 1 |
| | P | 1 | 9 | 9 | 1 | 1 | 1 | 1 | 9 | 1 | 2 | 1 | 1 |
| | F | 1 | 9 | 9 | 1 | 1 | 1 | 1 | 9 | 1 | 1 | 2 | 1 |
| | G | 1 | 9 | 9 | 1 | 1 | 1 | 1 | 1 | 1 | 2 | 1 | 1 |
| | H | 1 | 9 | 1 | 1 | 9 | 1 | 1 | 1 | 1 | 2 | 1 | 1 |
| | I | 1 | 9 | 9 | 1 | 1 | 1 | 1 | 1 | 9 | 2 | 1 | 1 |
| | J | 1 | 9 | 9 | 1 | 1 | 9 | 1 | 1 | 1 | 2 | 1 | 2 |
| | M | 1 | 9 | 9 | 1 | 1 | 1 | 1 | 1 | 1 | 2 | 1 | 2 |

### Molecular diversity indices

Allelic richness ($A_{Sc}$) was calculated for each population according to the rarefaction method (*El Mousadik & Petit, 1996*; *Kalinowski, 2004*) using Fstat software (*Goudet, 2013*). Estimates of $A_{Sc}$ were standardized to the smallest sample size ($n = 7$). Chlorotypic diversity ($H_{Sd}$) and its sampling variance were calculated according to the methodology given by *Nei (1987)* for each population separately and over the whole sample using Arlequin 3.11 (*Excoffier, Laval & Schneider, 2005*). To test for differences in allelic richness between the investigated geographical regions we employed a permutation test implemented in Fstat (*Goudet, 2013*) using 10,000 permutations of populations between geographical regions.

### Population genetic structure

To reveal structure in our dataset we used Spherikm (*Hill, Harrower & Preston, 2013*), software designed to analyse multivariate datasets by means of spherical $k$-means clustering (SKMC). The computations were based on a matrix of chlorotype frequencies in analysed populations. The statistically optimal number of clusters was assessed using the quasi-Akaike information criterion (*Hill, Harrower & Preston, 2013*). The partitioning of genetic variation within and between groups of populations identified by SKMC as well as between geographical regions was tested by analysis of molecular variance (AMOVA) using Arlequin 3.11 (*Excoffier, Laval & Schneider, 2005*). AMOVA computations were based on a distance matrix among the identified chlorotypes. We also carried out separate AMOVA analysis for populations within each geographical region in order to assess the strength of the separation between the regions.

### Distribution modelling

To reconstruct the potential distribution of *A. halleri* we used two palaeoclimate scenarios: mid-Holocene (ca. 6 kyr BP) and Last Glacial Maximum (LGM, ca. 22 kyr BP). Paleoclimatic data were obtained from simulations in the following Global Climate Models: CCSM4, MIROC-ESM and MPI-ESM-P. Bioclimatic variables calculated on the basis of these models and downscaled to 5 arc-minute resolution were downloaded from the WorldClim dataset (*Hijmans et al., 2005*) (http://www.worldclim.org), together with present-day climate data at the same resolution. We tested all the variables for multi-collinearity by examining the cross-correlations among them (Pearson's r) based on the 544 species occurrence records. Highly correlated variables ($r > 0.7$) were excluded from the models (*Dormann et al., 2013*), resulting in 8 variables representing temperature and precipitation: annual mean temperature (bio_1), mean diurnal temperature range (bio_2), isothermality (bio_3), temperature seasonality (bio_4), mean temperature of the wettest quarter (bio_8), mean temperature of the driest quarter (bio_9), annual precipitation (bio_12) and precipitation seasonality (bio_15). Areas covered by ice sheet (*Ehlers et al., 2004*) were excluded from the climatic layers of the LGM paleoclimate scenarios.

Distribution data were obtained from GBIF database (http://www.gbif.org/) as well as from our own field research carried out in France, Germany, Poland, Austria, Czech Republic, Slovakia and Ukraine. After initial screening for duplicates and data aggregation into a 5 min resolution raster we obtained 544 unique records that were used to calibrate

and validate the models (File S1). Data handling was done using GRASS GIS ver. 6.4 (http://grass.osgeo.org).

Seven different algorithms implemented in biomod2 ver. 3.1–48 (*Thuiller et al., 2009*; *Thuiller, 2014*) and MaxEnt ver. 3.3.3k (*Phillips, Anderson & Schapire, 2006*) were used: two regression methods (GLM—generalized linear models; GAM—generalized additive models), two classification methods (FDA—flexible discriminant analysis; CTA—classification tree analysis) and three machine-learning methods (GBM—generalized boosting model; RF—random forest for classification and regression and MAXENT—maximum entropy modeling). For each of the algorithms we ran 10 pseudo-absence replicates with 10,000 of pseudo-absences, to meet the minimum requirements of the algorithms used (*Barbet-Massin et al., 2012*). The models were fitted with 10 different random presence sets for each pseudo-absence run. This gave us a total of 100 replicates for each of the algorithms. Occurrence records were randomly divided into two subsets containing data for calibration (70%) and evaluation (30%) of models. We used the area under the receiver-operating characteristic (ROC) curve and true skill statistic (TSS) to evaluate model performance. These accuracy measures were calculated with reference to the current potential distribution only, due to the lack of independent and reliable fossil records for *A. halleri*.

Permutation procedure was used to define contributions of the variables to the models. In order to identify areas classified as suitable for species survival by the majority of algorithms (final consensus models) we performed ensemble forecasting (*Thuiller et al., 2009*). This procedure was used to eliminate the least reliable models (TSS < 0.7) and provided 7 ensemble models: mean of probabilities, coefficient of variation of probabilities, two models of confidence interval around the mean of probabilities, median of probabilities, models committee averaging (average of binary predictions) and weighted mean of probabilities. Binary transformation was carried out using a threshold that maximized the true skill statistic (TSS) to generate the most accurate predictions (*Jiménez-Valverde & Lobo, 2007*).

## RESULTS

### Chlorotype diversity

A total of 12 cpDNA haplotypes were found in the investigated populations (Table 4). All neighboring chlorotypes were linked by a single mutation (Fig. 1), except for G and H (separated by two mutations). Thus, the MST topology did not allow the division of chlorotypes into clearly demarcated groups separated by more than one mutation.

Chlorotypes did not show equal frequencies in overall sampling (Table 4). The most widely represented was chlorotype J, present in 26.4% of the samples analyzed. Chlorotypes E, F and G had a share of 15.47%, 12.19% and 12.11% respectively. The share of the remaining chlorotypes in the overall sampling was significantly lower than 10%.

Chlorotypes E and J were the most widespread geographically (Fig. 2), with their occurrence established respectively in 6 and 5 out of the 8 investigated geographical regions. Most of the haplotypes occupying tips and terminal branches of the MST were

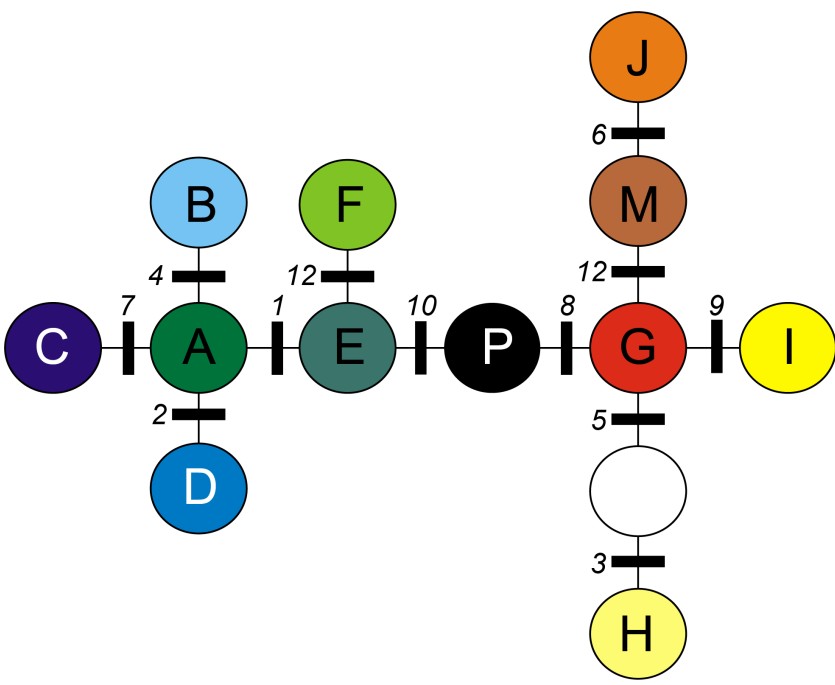

**Figure 1** **Minimum spanning tree (MST) presenting relationships between cpDNA haplotypes in *A. halleri*.** Coloured circles represent haplotypes, white circle represent missing haplotype. Numbers indicate mutations as given in Table 2.

more localized geographically. Chlorotypes B, C and H have been found only in the Bohemian Forest, chlorotype D in the Harz Mts., while chlorotype H only in the Eastern Carpathians (Fig. 2). Chlorotype G was found almost exclusively in the Eastern Carpathians, while chlorotype F showed a predominant occurrence in the Western Carpathians as well as in the geographically close region of the Northern Carpathian Foreland.

Genetic diversity indices: chlorotypic richness ($A_{Sc}$) and chlorotype diversity index ($H_{Sd}$) were calculated for each investigated population. They varied broadly from 1 to 4.983 for $A_{Sc}$ and from 0 to 0.893 for $H_{Sd}$ (Table 4). We also examined the geographical pattern of variation in chlorotypic richness. As shown in Fig. 3, populations with a high level of genetic diversity were co-located in the Bavarian Forest, the Harz Mts. and in the Western Carpathians (Tatra Mts.).

Also we compared genetic diversity indices between different geographical regions (Table 5). As expected, geographical regions differed substantially in terms of genetic diversity. Three regions: Western Carpathians, Harz Mts. and Bohemian Forest were found to be most diverse. The lowest genetic diversity was found in the Alps and in the Sudetes.

## Genetic structure

The clustering approach employed in the present study (spherical $k$-means clustering—SKMC) enabled us to study the structure present in our dataset on several levels. We examined a broad spectrum of different $k$ values from $k = 2$ to $k = 25$. The results of SKMC from $k = 2$ to $k = 10$ were plotted on the map (Fig. S2). Clearly, the populations

**Table 4 Chlorotype distribution among investigated populations of *A. halleri* and molecular diversity indices.**

| Pop | $n_i$ | A | B | C | D | Eα | E | F | G | H | I | J | K | L | M | O | $a_7$ | $H_{Sd}$ |
|---|---|---|---|---|---|---|---|---|---|---|---|---|---|---|---|---|---|---|
| A05 | 14 | . | . | . | . | . | 14 | . | . | . | . | . | . | . | . | . | 1.000 | 0.000 |
| A08 | 45 | . | . | . | . | . | 45 | . | . | . | . | . | . | . | . | . | 1.000 | 0.000 |
| A09 | 20 | . | . | . | . | . | 20 | . | . | . | . | . | . | . | . | . | 1.000 | 0.000 |
| CZ04 | 14 | . | 13 | . | . | . | . | . | . | . | 1 | . | . | . | . | . | 1.759 | 0.143 |
| CZ05 | 24 | . | 22 | . | . | . | . | 2 | . | . | . | . | . | . | . | . | 1.762 | 0.159 |
| CZ06 | 12 | . | 12 | . | . | . | . | . | . | . | . | . | . | . | . | . | 1.000 | 0.000 |
| CZ14 | 20 | . | . | . | . | . | . | . | . | . | . | 20 | . | . | . | . | 1.000 | 0.000 |
| CZ16 | 57 | . | . | . | . | . | 56 | . | . | . | . | 1 | . | . | . | . | 1.231 | 0.035 |
| CZ18 | 9 | . | . | . | . | . | 9 | . | . | . | . | . | . | . | . | . | 1.000 | 0.000 |
| CZ20 | 33 | . | . | . | . | . | . | . | . | . | 11 | 22 | . | . | . | . | 1.999 | 0.458 |
| CZ21 | 30 | . | . | . | . | . | . | . | . | . | 24 | 6 | . | . | . | . | 1.972 | 0.331 |
| CZ22 | 32 | . | . | . | . | . | . | . | . | . | . | 32 | . | . | . | . | 1.000 | 0.000 |
| D01 | 7 | . | . | 2 | . | . | 4 | . | . | . | . | 1 | . | . | . | . | 3.000 | 0.667 |
| D02 | 8 | 2 | . | 2 | . | . | 1 | . | . | 1 | . | 2 | . | . | . | . | 4.983 | 0.893 |
| D03 | 9 | 2 | . | . | . | . | 6 | . | . | 1 | . | . | . | . | . | . | 2.960 | 0.556 |
| D04 | 11 | 5 | . | . | . | . | 2 | . | . | 4 | . | . | . | . | . | . | 2.990 | 0.691 |
| D08 | 12 | 1 | . | . | 10 | . | 1 | . | . | . | . | . | . | . | . | . | 2.674 | 0.318 |
| D09 | 18 | 3 | . | . | 15 | . | . | . | . | . | . | . | . | . | . | . | 1.962 | 0.294 |
| D11 | 18 | 5 | . | . | 9 | . | 4 | . | . | . | . | . | . | . | . | . | 2.987 | 0.660 |
| D12 | 18 | 4 | . | . | 6 | . | 8 | . | . | . | . | . | . | . | . | . | 2.989 | 0.680 |
| D13 | 20 | . | . | . | 17 | . | 3 | . | . | . | . | . | . | . | . | . | 1.940 | 0.268 |
| D14 | 11 | 1 | . | . | 10 | . | . | . | . | . | . | . | . | . | . | . | 1.879 | 0.182 |
| PL02 | 21 | . | . | . | . | . | . | 14 | . | . | . | 7 | . | . | . | . | 1.999 | 0.467 |
| PL03 | 15 | . | . | . | . | . | . | 13 | . | . | . | 2 | . | . | . | . | 1.934 | 0.248 |
| PL07 | 19 | . | . | . | . | . | 4 | 15 | . | . | . | . | . | . | . | . | 1.985 | 0.351 |
| PL08 | 12 | . | . | . | . | . | . | 5 | . | . | . | 7 | . | . | . | . | 2.000 | 0.530 |
| PL32 | 27 | . | . | . | . | . | 11 | 9 | . | . | 7 | . | . | . | . | . | 2.992 | 0.681 |
| PL33 | 21 | . | . | . | . | . | . | 7 | . | . | 14 | . | . | . | . | . | 1.999 | 0.467 |
| PL37 | 40 | . | . | . | . | . | . | . | . | . | . | 40 | . | . | . | . | 1.000 | 0.000 |
| PL38 | 29 | . | . | . | . | . | . | . | . | . | . | 29 | . | . | . | . | 1.000 | 0.000 |
| PL39 | 39 | . | . | . | . | . | . | . | . | . | . | 39 | . | . | . | . | 1.000 | 0.000 |
| PL40 | 39 | . | . | . | . | . | . | . | . | . | . | 39 | . | . | . | . | 1.000 | 0.000 |
| PL41 | 39 | . | . | . | . | . | . | . | . | . | 1 | 38 | . | . | . | . | 1.329 | 0.051 |
| PL42 | 39 | . | . | . | . | . | . | . | . | . | 28 | 11 | . | . | . | . | 1.994 | 0.416 |
| PL43 | 39 | . | . | . | . | . | . | . | . | . | . | 39 | . | . | . | . | 1.000 | 0.000 |
| PL44 | 41 | 14 | . | 27 | . | . | . | . | . | . | . | . | . | . | . | . | 1.999 | 0.461 |
| PL45 | 39 | . | . | . | . | . | 2 | 36 | . | . | . | 1 | . | . | . | . | 1.883 | 0.148 |
| PL46 | 23 | . | . | . | . | . | . | . | 23 | . | . | . | . | . | . | . | 1.000 | 0.000 |
| PL47 | 24 | . | . | . | . | 4 | . | . | 20 | . | . | . | . | . | . | . | 1.952 | 0.290 |
| PL48 | 20 | . | . | . | . | . | . | . | 20 | . | . | . | . | . | . | . | 1.000 | 0.000 |

*(continued on next page)*

**Table 4** (*continued*)

| Pop | $n_i$ | A | B | C | D | E$\alpha$ | E | F | G | H | I | J | K | L | M | O | $a_7$ | $H_{Sd}$ |
|-----|-------|---|---|---|---|-----------|---|---|---|---|---|---|---|---|---|---|-------|----------|
| | | | | | | | | | | | | | | | | | Chlorotype | |
| PL49 | 22 | . | . | . | . | 2 | . | . | 20 | . | . | . | . | . | . | . | 1.798 | 0.173 |
| PL50 | 34 | . | . | 34 | . | . | . | . | . | . | . | . | . | . | . | . | 1.000 | 0.000 |
| SK02 | 22 | . | . | . | . | . | 4 | 18 | . | . | . | . | . | . | . | . | 1.967 | 0.312 |
| SK05 | 46 | . | . | . | . | . | . | 19 | 1 | . | 25 | 1 | . | . | . | . | 2.565 | 0.545 |
| SK10 | 21 | . | . | . | . | 7 | 2 | 11 | . | . | . | 1 | . | . | . | . | 3.377 | 0.633 |
| SK12 | 16 | . | . | . | . | 7 | 2 | 7 | . | . | . | . | . | . | . | . | 2.915 | 0.642 |
| UA01 | 31 | . | . | . | . | 21 | . | . | 10 | . | . | . | . | . | . | . | 1.998 | 0.452 |
| UA02 | 32 | . | . | . | . | 30 | . | . | 2 | . | . | . | . | . | . | . | 1.638 | 0.121 |
| UA04 | 32 | . | . | . | . | 5 | . | . | 27 | . | . | . | . | . | . | . | 1.932 | 0.272 |
| UA05 | 32 | . | . | . | . | 6 | . | . | 14 | . | . | . | . | . | 12 | . | 2.962 | 0.653 |
| UA06 | 16 | . | . | . | . | 1 | . | . | 12 | . | . | . | . | . | 3 | . | 2.671 | 0.425 |
| UA07 | 8 | . | . | . | . | . | . | . | 6 | . | . | . | . | . | 2 | . | 2.000 | 0.429 |
| Total | 1,280 | 37 | 47 | 65 | 67 | 83 | 198 | 156 | 155 | 6 | 111 | 338 | 0 | 0 | 17 | 0 | 6.927 | – |

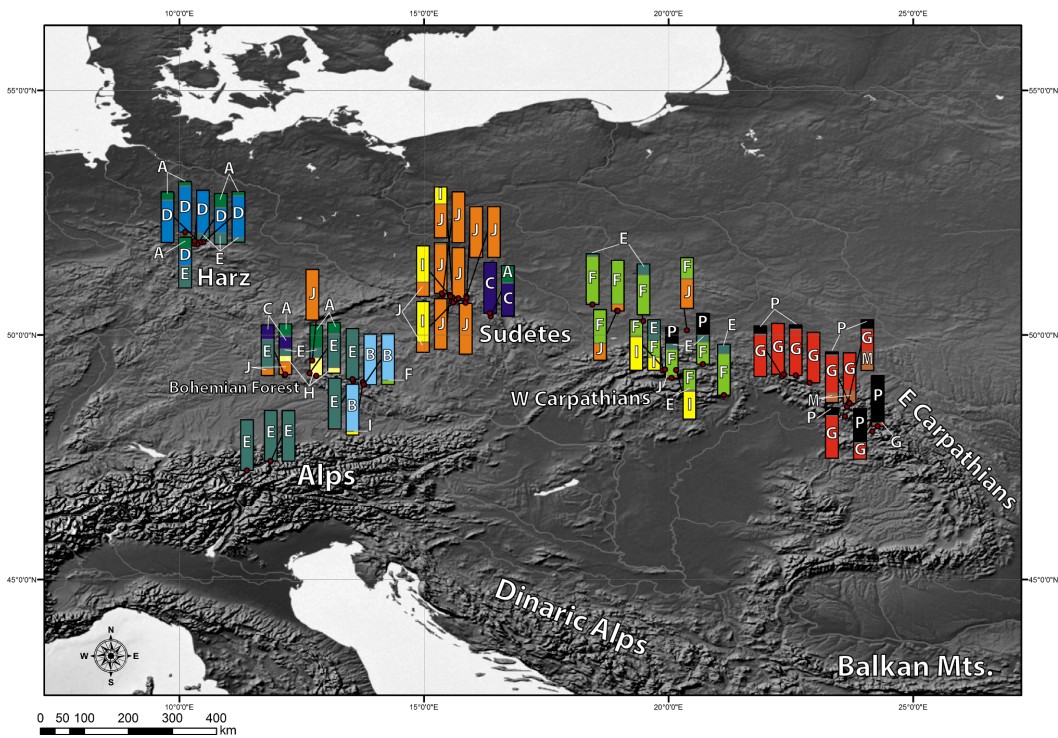

**Figure 2** **Geographic distribution of cpDNA haplotypes present in the investigated populations of *A. halleri*.** Bar charts represent the frequency of each haplotype in each investigated population.

from Eastern Carpathians formed one stable cluster (present in all the *k* levels), that differed from all the other populations. A subdivision of the populations into 6 clusters (Fig. 4) was statistically optimal (Fig. 5A) and had a high support in AMOVA (Fig. 5B). We carried out separate AMOVA analyses to test the distribution of genetic variance within and between

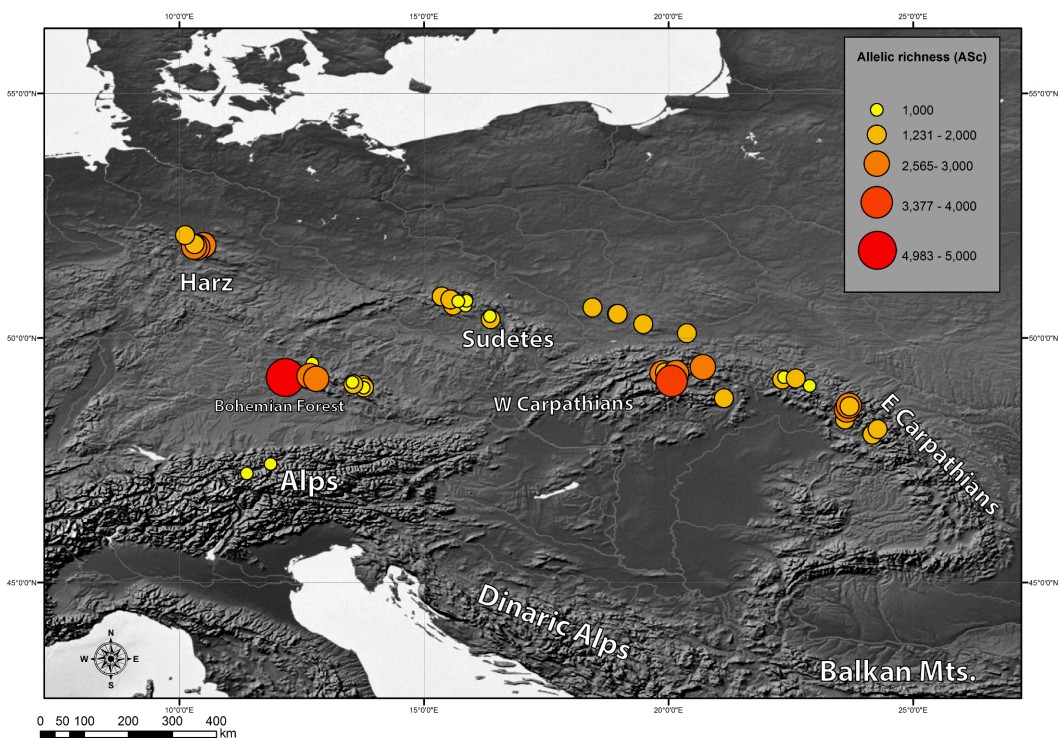

**Figure 3  Geographic distribution of within-population cpDNA allelic richness ($A_{Sc}$) in the populations investigated.**

**Table 5  Comparison of within-population diversity indices among different geographical regions.**

| Geographical group | n | $A_{Sc}$ | $H_{Sd}$ |
|---|---|---|---|
| Western Carpathians | 5 | 2.636 | 0.547 |
| Eastern Carpathians | 11 | 1.895 | 0.285 |
| Sudetes | 12 | 1.358 | 0.143 |
| Harz | 6 | 2.405 | 0.420 |
| Alps | 3 | 1.000 | 0.000 |
| Bohemian Forest | 10 | 2.169 | 0.181 |
| N Carpathian Foreland | 5 | 1.960 | 0.303 |

**Notes.**
$n$, Number of populations in a region; $A_{Sc}$, Chlorotypic richnes; $H_{Sd}$, Chlorotype diversity index.

groups identified by SKMC. Results of SKMC evidenced the presence of some level of genetic admixture in almost all studied geographical regions except in the Harz Mts., the Alps, and the Eastern Carpathians.

AMOVA performed without grouping populations showed that 79.16% of the total genetic variation was found between populations (Table 6). When assuming six groups of populations (according to $k$-means clustering), as much as 65.8% of the total variation was observed between groups of populations, whereas 15.82% was found among populations within groups (Table 6). These percentages of genetic variation were 42.88% and 37.90%,
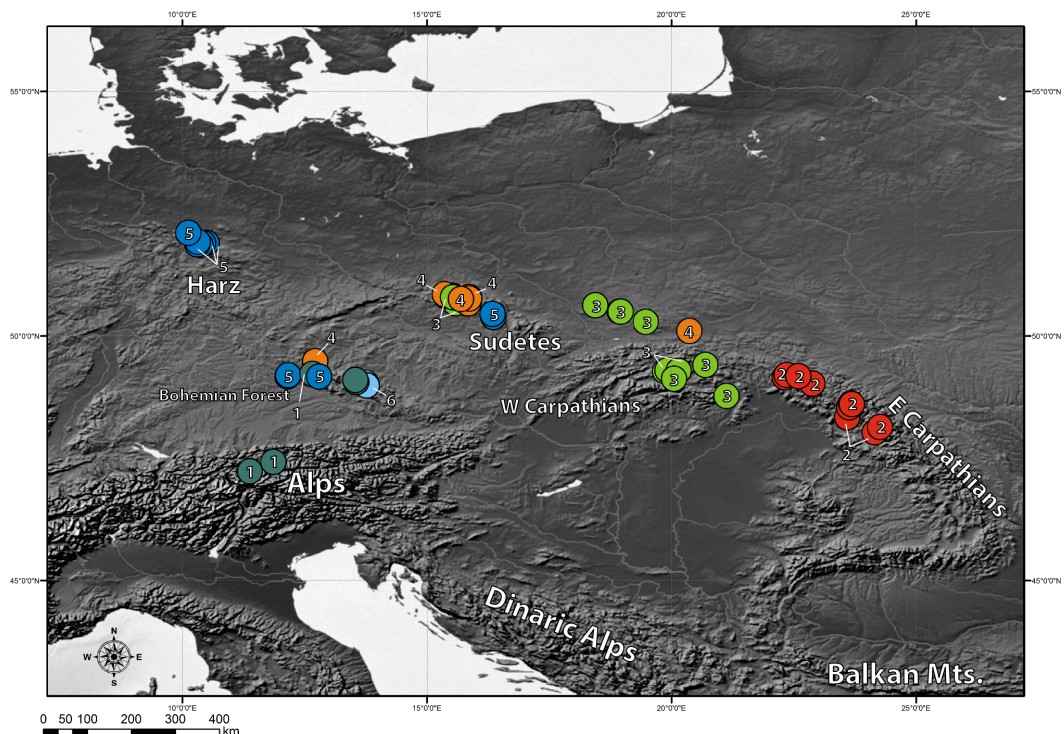

**Figure 4** **Results of spherical *k*-means clustering (SKMC).** SKMC of investigated populations carried out using Shperikm (*Hill, Harrower & Preston, 2013*). Statistically optimal level of *k* = 6 was presented (see Fig. S2 for other levels of *k*). Different colours (and numbers from 1 to 6) correspond to different clusters identified by SKMC. Population differentiation was inferred from a data matrix of chlorotype frequencies.

respectively, when assuming seven groups (characterised according to the geographical regions sampled; Table 6). Separate AMOVAs performed within each investigated geographical region revealed the highest percentage of among-population variation in the Sudetes and the Bohemian Forest: 85.66% and 74.57%, respectively (Table 6). The lowest values of among-population variation was found in the Alps, where we recorded the presence of just one chlorotype (0%; Table 6), and in the Harz Mts. (18.55%; Table 6). All the remaining geographic regions showed intermediate level of among-population variation.

## Distribution modelling

Model performance was assessed using two different statistics: True Skill Statistic (TSS) and Area under Receiver Operating Characteristic (ROC). All models performed well and had TSS >0.7 and ROC >0.88. The performance of two models CTA and FDA was weaker than the performance of the remaining methods, but still of acceptable quality.

Two (CCSM4, MPI ESM) out of the three climatic models suggest that the species survived the LGM (ca. 22 kyr BP) only in the southernmost part of the Carpathians (Fig. 6). The Dinaric Alps and Balkan Mts. were also among the potential areas for species survival in southern Europe during the LGM (Fig. 6). Vast areas north from the Alps can be also considered as suitable for the survival of *A. halleri* (Fig. 6).

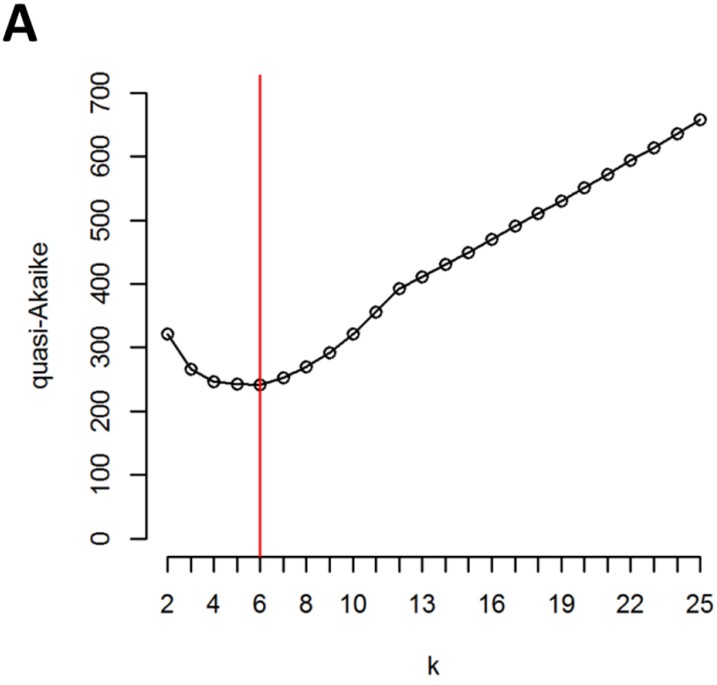

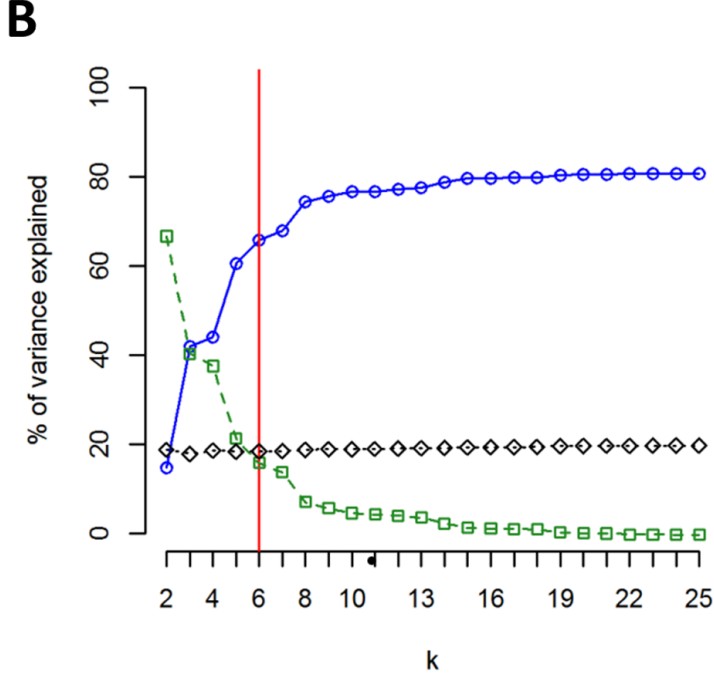

**Figure 5   Results of SKMC and AMOVA.** (A) Value of the quasi-Akaike information criterion as a function of *k* (number of groups identified by SKMC analysis). Statistically optimal solution (having the lowest value of the quasi-Akaike criterion) is marked with a red line. (B) Percent of genetic variance among groups of populations identified by SKMC (blue), among populations within groups (green) and within populations (black) calculated by AMOVA. Statistically optimal solution of SKMC is marked with a red line.

**Table 6  Results of AMOVA analysis.** All results were significant with $\alpha = 0.05$. Significance tests were carried out using permutation test (10,100 permutations).

| | Source of variation | d.f. | Sum of squares | Variance components | Precentage of variation |
|---|---|---|---|---|---|
| All populations | Among populations | 51 | 1329.890 | 1.09838 | 79.16 |
| | Within populations | 1,233 | 356.479 | 0.28912 | 20.84 |
| Western Carpathians | Among populations | 5 | 33.292 | 0.23835 | 23.95 |
| | Within populations | 147 | 111.244 | 0.75676 | 76.05 |
| Eastern Carpathians | Among populations | 9 | 32.980 | 0.14726 | 46.44 |
| | Within populations | 230 | 39.066 | 0.16985 | 53.56 |
| Sudetes | Among populations | 11 | 408.052 | 1.01023 | 85.66 |
| | Within populations | 427 | 72.235 | 0.16917 | 14.34 |
| Harz | Among populations | 5 | 6.934 | 0.06793 | 18.55 |
| | Within populations | 91 | 27.148 | 0.29833 | 81.45 |
| Alps | Among populations | 2 | 0.000 | 0.00000 | 0.00 |
| | Within populations | 76 | 0.000 | 0.00000 | 0.00 |
| Bohemian forest | Among populations | 9 | 133.130 | 0.91778 | 74.57 |
| | Within populations | 161 | 50.379 | 0.31291 | 25.43 |
| N Carpathian Foreland | Among populations | 4 | 18.696 | 0.20423 | 26.78 |
| | Within populations | 101 | 56.408 | 0.55849 | 73.22 |
| $k$-means clustering (6) | Among groups | 5 | 1102.418 | 1.03618 | 65.82 |
| | Among populations | 46 | 290.472 | 0.24899 | 15.82 |
| | Within populations | 1,233 | 356.479 | 0.28912 | 18.36 |
| Geographic location (7) | Among groups | 6 | 759.806 | 0.64481 | 42.88 |
| | Among populations | 45 | 633.085 | 0.56992 | 37.90 |
| | Within populations | 1,233 | 356.749 | 0.28912 | 19.23 |

We also analysed the potential distribution of *A. halleri* during the Holocene climatic optimum (ca. 6 kyr BP). Our models showed that recolonization must have advanced very slowly in the Carpathians, when compared with areas north of the Alps. All the models showed that conditions facilitating the spread of *A. halleri* occurred much earlier in between the Western Carpathians, the Sudetes and areas north from the Alps, while suitable areas in the Eastern Carpathians were at first much more restricted (Fig. 6).

## DISCUSSION

### Carpathian populations of *A. halleri*

Our data suggest a clear differentiation among populations from western and eastern part of the Carpathians. Those population groups differ in terms of chlorotype composition and frequencies. The same differentiation also appears clearly in the SKMC analysis since the western and eastern populations were grouped in different clusters from $k = 2$. This pattern of genetic variation seems to follow the division between the eastern and western part of the Carpathians which was first recognized by *Wołoszczak (1896)* and was established on the basis of floristic data. The nature of the barrier between the western and eastern part of

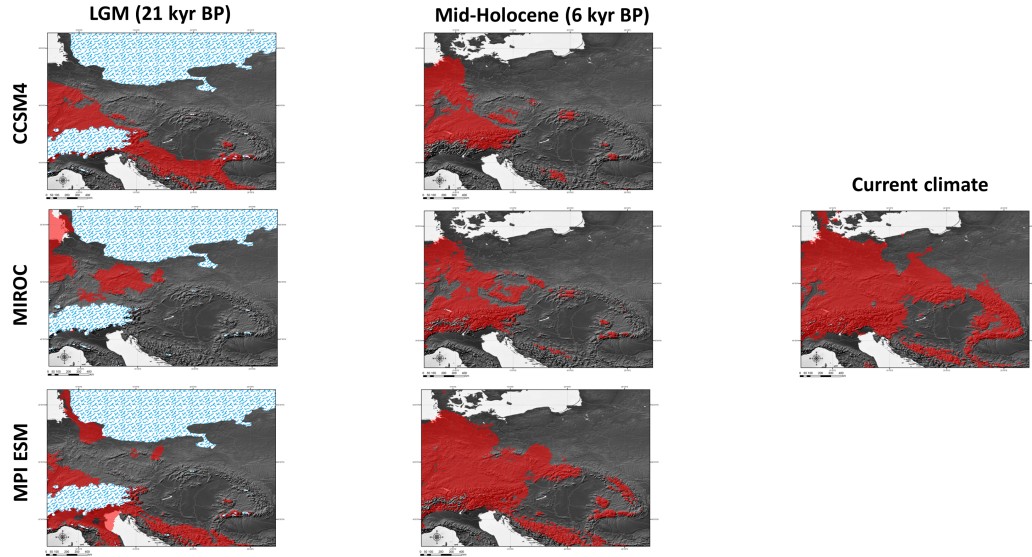

**Figure 6 Results of modelling experiments.** Binary maps of distributions based on the results of ensemble models (mean of probabilities) are shown. Results are based on the data from three different paleoclimate models: CCSM4, MIROC and MPI ESM, as well as current climate observations. Species range was reconstructed for two time periods: Last Glacial Maximum (LGM, ca. 21 kyr BP) and Mid-Holocene (ca. 6 kyr BP).

Carpathians has been the subject of many studies employing different methodologies from floristic (*Pax, 1898*; *Jasiewicz, 1965*) to cytologic (*Mráz & Szelag, 2004*) and genetic (*Mráz et al., 2007*; *Ronikier, Cieślak & Korbecka, 2008a*; *Těšitel et al., 2009*). It has been hypothesized that specific climatic and orographic conditions of the westernmost part of Bieszczady Mts. (also known as Bukovske Vrchy Mountains) are among the main factors influencing the genetic landscape of this part of Carpathians (*Domin, 1940*). It seems that results of our study may suggest that differentiation between western and eastern part of Carpathians may be a historical phenomenon connected with recolonisation of the area by plants that survived in different refugia. In this case we do not need to postulate the presence of a specific barrier responsible for the existence of genetic discontinuity between western and eastern part of Carpathians since this phenomenon could be also explained by colonisation from two different directions together with the presence of gene flow between the two groups as was evidenced by our study (i.e., the presence of haplotype P in western and eastern Carpathians).

The Eastern Carpathian populations are characterized by the presence of haplotype G, at very high frequencies. This haplotype was considered as ancestral by *Pauwels et al. (2005)*. The presence of the ancestral haplotype G at high frequencies has also been recorded in populations located south of the Alps as well as in the south-eastern part of this mountain range (*Pauwels et al., 2012*). Our genetic data showing low genetic diversity in the Eastern Carpathians suggest relatively recent recolonisation of this area by *A. halleri* as was also indicated by the modelling experiments. This is surprising, when we take into account that several studies have demonstrated (with high probablity) that a glacial refugium existed in the Eastern Carpathians (*Willis & Van Andel, 2004*; *Tollefsrud et al., 2008*). The low genetic

diversity in the Eastern Carpathians was, however, also confirmed by *Tollefsrud et al. (2008)* in *Picea abies*. These authors suggested a genetic bottleneck as a result of substantial decrease in population size as a reason of this phenomenon. The same scenario cannot be ruled out for *A. halleri*, although modelling experiments carried out during the present study suggest postglacial recolonisation as a more plausible explanation of low genetic diversity in the Eastern Carpathians.

The Western Carpathian populations are characterized by high levels of genetic diversity and the presence of a private haplotype F at high frequencies. Both premises suggest the possibility of *in situ* glacial survival of the species in the Western Carpathians.

Survival of plant species in the Western Carpathians during the LGM has been a focal point of many studies employing different methodologies. These works have shown that the existence of a Western Carpathian refugium is quite probable for many plant species, including mountain plants. Macrofossil charcoal fragments found in Kraków, just about 30 km in a straight line from population PL8 and about 40 km from PL 7, indicate full glacial presence of *Pinus*, *Larix* and *Abies* from 26 to 28 kyr BP—during the coldest period of LGM that spanned between ca. 36–16 kyr BP (*Damblon, Haesaerts & Van der Plicht, 1996*; *Willis & Van Andel, 2004*; *Musil, 2003*). Dates from humic soil further down in the sequence are even earlier, and indicate the presence of trees as early as 36 kyr BP (*Willis & Van Andel, 2004*). There is also taxonomic evidence supporting the hypothesis of longstanding survival of different plant species in the Western Carpathians. *Saxifraga wahlenbergii* Ball. and *Delphinium oxysepalum* Borb. et Pax are good examples here. These two species are endemic to the Western Carpathians and occupy isolated systematic positions, what suggests that they are both of Tertiary age (*Mirek & Piekos-Mirkowa, 1992*). The survival of common yew (*Taxus baccata*), was also documented in charcoal for Moravany in Slovakia with a radiocarbon date of ca. 18 kyr BP (*Lityńska-Zając, 1995*). There is also a large body of phylogeographic evidence that indicates the existence of a major northern refugium for a variety of animal taxa in the area around the Carpathians, with some lineages predating LGM (*Provan & Bennett, 2008*).

On the other hand, results of niche modelling conducted during the present study suggest postglacial recolonization of the area. In this scenario high genetic diversity in the area could be at least partially explained by relatively recent gene flow (occurring later than 6 kyr BP, according to our modelling experiments) from the Eastern Carpathians. The presence of a genetic admixture in Western Carpathian populations was also revealed by *Pauwels et al. (2012)* in *A. halleri* and by *Těšitel et al. (2009)* in *Melampurum sylvaticum*. It seems that the question of the existence of glacial refugium for *A. halleri* in the Western Carpathians should remain open taking into account that evidence are still not conclusive.

In the SKMC analysis, populations from the Western Carpathians were clustered together with populations from upland regions of southern Poland. This fact supports the hypothesis of a Western Carpathian origin of populations located north of the Western Carpathians. However, for the reasons stated above, it is very difficult to say when these populations were founded. We have shown that populations of *A. halleri* in western and eastern part of Carpathians form two different genetic groups. It might be hypothesized that these groups could be derived from two different refugia. Populations in the Eastern

Carpathians might belong to the group that survived in a southern refugium (areas south of the Alps, Dinaric Alps and Balkan Mts.) as suggested by *Pauwels et al. (2012)*.

## Genetic differentiation between the Harz, the Bohemian Forest and the Alps

The situation in the areas north and north-east of the Alps is more complex than in the Carpathians. SKMC analysis showed that at least two groups of populations can be recognized regardless of the level of k: populations from the Harz Mountains and from the Alps. Populations from the Bohemian Forest were usually assigned to different clusters, forming very heterogeneous group. This heterogeneity was also evidenced in AMOVA.

Differentiation is also apparent between the regions of the Harz Mts. and the Bohemian Forest, with the first harbouring haplotype D, which is not present in the Bohemian Forest. In the latter region the occurrence of haplotype C can be observed, which, in turn, is present neither in the Harz, nor in the Alps or the Šumava. A relatively recent (postglacial) origin of this differentiation could be hypothesized as both haplotypes occupy external nodes of the MST tree.

It is not easy to explain this pattern of genetic differentiation. Some ideas might be provided by the results published by *Tollefsrud et al. (2008)*. Investigating the genetic variation of Norway Spruce together with pollen data they established that one possible glacial refugium of the species might have extended from the northern slopes of the Alps up to the Šumava (the Bohemian Massif). It seems that *A. halleri* might have also survived in a vast area, and that its Pleistocene distribution covered not only the region mentioned above, but also extended northwards and westwards to the Ardennes and Hautes Fagnes (High Fens) in Belgium. So far one natural population from this area has been tested by *Pauwels et al. (2008)*. This, nowadays isolated, population from Hautes Fagnes, harbouring a chlorotype with an extremely restricted geographical range (*Pauwels et al., 2008*), might be the trace of a past *A. halleri* distribution in Western Europe. The vast extent of possible refugial areas north of the Alps was also clearly evidenced by our modelling experiments. Other studies based on molecular methods also suggest the presence of a glacial refugium in the Central Europe (*Reisch, Poschlod & Wingender, 2003*; *Koch, 2002*; *Rejzková et al., 2008*).

Genetic variation in the closely related species *Arabidopsis lyrata* from the Harz, southern Germany and the Alps (*Clauss & Mitchell-Olds, 2006*), can give us some insights into a possible explanation of this pattern. Studies on genetic variation of *A. lyrata* (carried out on the basis of nuclear microsatellite loci) showed high within-population diversity throughout central Europe, accompanied by low regional differentiation and geographically widespread polymorphism. The authors hypothesized that (given the unlikeliness of gene flow) a common gene pool must have existed for central European populations (*Koch & Matschinger, 2007*). It should be noted that "central European" in this case is not a precise term and describes sites located approximately between the 10th and 16th eastern meridian. This area corresponds to the locations of the populations sampled by us north of the Alps in the the Bohemian Forest and the Harz. The same scenario is also probable for *A. halleri*, where haplotype E, present in all regions north of the Alps, can be interpreted as a testimony

to a common gene pool in the past. Other haplotypes, with restricted geographical range, would be, under this model, local derivatives that evolved after climatic warming and fragmentation of a previously vast range.

The Bohemian Forest hosts a population with the highest genetic diversity in the whole sampled area. The question whether this high genetic diversity should be attributed to the existence of a glacial refugium in this region or the presence of a contact zone between eastern and western lineages remains open. We think that results cited above (*Clauss & Mitchell-Olds, 2006*; *Tollefsrud et al., 2008*) as well as data clearly suggesting the presence of a glacial refugium for *Fagus* in the southern part of Czech Republic (*Magri et al., 2006*) make the hypothesis of the existence of a glacial refugium in this area more probable. The fact that even nowadays the range of *A. halleri* and Norway Spruce in Central Europe is highly similar (*Tollefsrud et al., 2008*) also suggest this scenario. However, no palaeobotanical data are available for *A. halleri*. Therefore, such a hypothesis cannot be fully confirmed or rejected. It should be also noted that the area between the 10th and 16th eastern meridian has been identified as a contact zone for some plant species (*Fjellheim et al., 2006*; *Daneck et al., 2011*). It is clear that further detailed studies in this region, also including other taxa, are needed to clarify these findings more precisely.

### Origin of *A. halleri* populations in the Sudetes

We have shown that *A. halleri* populations in Sudetes are characterized by a very low genetic variation, with most of the populations harbouring only one cpDNA haplotype. This finding is not surprising, given the evidence from geological research showing that this region was severely impacted during the glacial period. Is seems that at least one glacial maximum (48–43 kyr BP) had a devastating effect on the regional flora, mainly due to the close proximity of the continental ice sheet (*Marks, 2005*). Geological evidence show that even during LGM this region was affected by the presence of the mountain glaciers (*Badura & Przybylski, 1998*). Therefore, it has been postulated that *in situ* glacial refugia, supporting the remains of the autochthonous flora, did not exist in the Sudetes (*Mitka et al., 2007*). Quite a different scenario involving survival on nunataks and in peripheral refugia has been suggested for the Alps (*Stehlik, 2000*). It is noteworthy that even nowadays the climate of the highest parts of the Sudetes is cold enough to maintain the occurrence of tundra-like ecosystems (*Soukupova et al., 1995*) . It seems therefore, that *A. halleri* populations in the Sudetes could be of a recent (postglacial) origin. This is supported by the very low genetic variation found in Sudetes as well as the fact that despite dense sampling, we have not found any chlorotypes specific to this area. On the basis of the evidence from cpDNA variation, it could be hypothesized that populations from the Bohemian Forest could be a source of migrants that established new populations of the species in this region after LGM.

We have also found the presence of haplotype I in high-mountain populations of *A. halleri* in the Sudetes. The occurrence of this uncommon haplotype has also been recorded in high-mountain populations in the Western Carpathians as well as for one population in the Bohemian Forest. A similar pattern have been also recognized in *Pulsatilla vernalis* (*Ronikier et al., 2008b*). The most reasonable explanation for the observed pattern is the assumption that a contact between the Sudetic and Carpathian flora occurred in

the past. *Mitka et al. (2007)* suggested that this contact could have occurred especially for high-mountain taxa, which could easily disperse within the open landscapes that were present between the Sudetes and the Carpathians during glacial maxima. This supposition is also supported by the floristic evidence showing that several high-mountain taxa such as *Erigeron macrophyllus* Herbich, *Melampyrum herbichii* Woł., *Sesleria tatrae* (Degen) Deyl and *Thymus carpathicus* Čelak. that are present in the Carpathians, occur also in high-mountain environments of the Sudetes (*Pawłowski, 1969*). The connections between the Carpathian and Sudetic populations of *A. halleri* surely require further studies.

## Taxonomic concept of *A. halleri* in the Carpathians

Taxonomy of *A. halleri* is still much debated and to date three European subspecies have been recognised: subsp. *halleri*, subsp. *tatrica* (Pawł.) Kolník, and subsp. *dacica* (Heuff.) Kolník (*Hohmann et al., 2014*). This division is based on a morphological study of the Carpathian populations published by *Kolník & Marhold (2006)*. Our results seem to question this taxonomic division that has already been cited by various authors (*Clauss & Mitchell-Olds, 2006*; *Koch, Wernisch & Schmickl, 2008*). We have shown that main genetic groups identified by cpDNA variation are not consistent with the division proposed by *Kolník & Marhold (2006)*, nor with the distribution of the described taxa.

## CONCLUSIONS

1. Genetic variation in *A. halleri* is strongly geographically structured within the investigated area and at least 6 clusters of populations can be identified on the basis of cpDNA variation.
2. There is a clear genetic differentiation within the Carpathian populations. Two distinct clusters of populations were identified in western and eastern part of the mountain range. It is clear that the two groups originated from two different glacial refugia. There are traces of gene flow between the two groups.
3. The possibility of the existence of a glacial refugium in the Western Carpathians and the Bohemian Forest cannot be ruled out on the basis of the present research but the evidence is not conclusive.
4. It seems that the area of Sudetes was colonised after LGM. Populations from the Bohemian Forest can be hypothesised as a source of the migrants which established *A. halleri* populations in the area.

## ACKNOWLEDGEMENTS

The authors would like to thank Prof. Ian C. Trueman (University of Wolverhampton, UK) for improving the English of the manuscript. Krkonošský národní park (CZ), Karkonoski Park Narodowy (PL), Tatrzański Park Narodowy (PL), Bieszczadzki Park Narodowy (PL) and Karpatskij biosfernij Zapovidnik (UA) are acknowledged for granting permits to collect plant samples within protected areas of these national parks.

### Funding
The study was founded by grants from: Polish Ministry of Science and Higher Education N303415336 and Égide (Campus France) 7298/R08/R09. The funders had no role in study design, data collection and analysis, decision to publish, or preparation of the manuscript.

### Grant Disclosures
The following grant information was disclosed by the authors:
Polish Ministry of Science and Higher Education: N303415336.
Égide (Campus France): 7298/R08/R09.

### Competing Interests
The authors declare they are no competing interests.

### Author Contributions
- Pawel Wasowicz conceived and designed the experiments, performed the experiments, analyzed the data, contributed reagents/materials/analysis tools, wrote the paper, prepared figures and/or tables, reviewed drafts of the paper.
- Maxime Pauwels conceived and designed the experiments, performed the experiments, contributed reagents/materials/analysis tools, wrote the paper, reviewed drafts of the paper.
- Andrzej Pasierbinski conceived and designed the experiments, performed the experiments, analyzed the data, contributed reagents/materials/analysis tools, prepared figures and/or tables, reviewed drafts of the paper.
- Ewa M. Przedpelska-Wasowicz performed the experiments, analyzed the data, contributed reagents/materials/analysis tools, prepared figures and/or tables, reviewed drafts of the paper.
- Alicja A. Babst-Kostecka performed the experiments, contributed reagents/materials/-analysis tools, reviewed drafts of the paper.
- Pierre Saumitou-Laprade and Adam Rostanski conceived and designed the experiments, performed the experiments, contributed reagents/materials/analysis tools, reviewed drafts of the paper.

### Field Study Permissions
The following information was supplied relating to field study approvals (i.e., approving body and any reference numbers):

Permits for plant sampling were obtained from the following national parks: Krkonossky narodni park (CZ, permit no. 01004/2009), Karkonoski Park Narodowy (PL permits no. 24/2007, 2/2008)/, Tatrzanski Park Narodowy (PL, permit no. NB-056/119/07), Bieszczadzki Park Narodowy (PL, permits no. 54/06, 40/07) and Karpatskij biosfernij Zapovidnik (UA, permit no. 01110607/07).

## Data Availability

Raw data—chlorotype frequencies in each population—can be found in Table 4.

## Supplemental Information

Supplemental information for this article can be found online at http://dx.doi.org/10.7717/peerj.1645#supplemental-information.

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
