# Peer review of "Phylogeography of Arabidopsis halleri (Brassicaceae) in mountain regions of Central Europe inferred from cpDNA variation and ecological niche modelling"

_PeerJ, doi:10.7717/peerj.1645_

## Round 0.1 · original submission · Major Revisions

All three reviewers find merit in the article, and I believe a sufficiently revised manuscript would be suitable for publication in PeerJ. The reviewers do raise a number of concerns about methods and writing, however, that will need to be addressed.

Reviewer 1 ·

Basic reporting

No comments

Experimental design

No comments

Validity of the findings

No comments

Additional comments

Comments to the Author:
This paper by Wasowicz et al. tackles the problem of phylogeography of Arabidopsis halleri by using the DNA markers from chloroplast genome. The main contribution of the paper is 1) to provide evidence of northern refugium besides the frequently reported southern refugium in Europe 2) to shed light on the phylogeographic history of plants occurring in Central Europe. Besides the traditional analysis (population structure and phylogenetic trees reconstruction) in phylogeographic studies, the up-to-date niche modeling is a good add-to to the manuscript. Overall, the paper is well written, and the question was clearly addressed, answered and discussed. I think this paper is fit for the journal, but still need some polishment.

I have some specific comments, which I list below:

Format and expression questions:
Page 2, line 11: north and northeast of Alps --→ at north and northeast of Alps
Page 3, line 7: locate → located
Page 3, line 17: the mountain plants → the mountains
Page 4, line 15: remove the “3” in “patterns 3”;
Page 6, line 5: change “Genotyping procedure” as “Genotyping procedure:”
Page 6, line 6, 7: the name of the gene should be italic; and please all of them as the same format;
Page 12, line 9: change “We examined also geographical” as “We also examined geographical”;
Page31, Table 1: the expression of latitude and longitude need to be improved as “ 47o13’46.68’’ ”;
Page33. Table 2: the name of the species should be changed as “A. halleri”; also for the following tables;

Introduction part:
For the species parts, it is probably good to describe more details about this species, for example, some information about its relatives within the same range, and whether it could be hybridized with other relatives. It would be helpful for the discussion, because only cp markers were used to resolve its history.
Why only choosing the marker from cpDNA? What’s the advantage and disadvantage?

I would like to have a clear clue about the sampling, like the number of populations, and the number of individuals in each population.

Methods part:
It will be help the reader to understand if the author can write the rationale for the three different genotyping methods used in the manuscript.


Results and Discussion:
Only using the markers from the chloroplast genome is really limited to discuss lots of details. As the chloroplast markers are inherited from single parent, it may not represent the real phylogeographic history of the species. It will be good if the author can point it out.
Based solely on the diversity and frequency of cp markers to discuss some taxonomic questions is not credible. Please list some limitations or add some other reference.

Reviewer 2 ·

Basic reporting

The language is sometimes awkward, or inefficient to the point of being confusing. After reading very closely, I would say this is inherently solid research, however, it reads flat because of awkward wording, wandering paragraphs, and vague aims.

I have noted most sections that are not well-worded so that the authors can improve the text.

The introduction does not adequately situate the article, and the reader doesn't fully understand the point of the work until reading the discussion. There is only a passing mention to the hypothesis of northern refugia in the introduction. However, in the discussion, we find out that the work makes a significant contribution to scientific debate about the taxonomy of the A. halleri, where glacial refugia could or could not have been in central europe, and the contribution of northern refugia to present day ranges.

I will also suggest a few improvements to the figures, although the figures do meet the requirements as they are:
Figure 2: It is difficult to figure out what the bars lines and points mean and what geographic region the bars are actually associated with. Can you replace the bars/points/lines with one tiny pie-chart for each population?
Figure 4: I don't think all the k groups you present are really necessary to understand the results. Perhaps k=6 and k=2 only? or cut it at k=6? Only k=2 (simplest) and k=6 (statistically optimal) are discussed in the text.
Another figure: A schematic map figure showing the possible refugia and migration routes proposed and evaluated by the authors would help with clarity. It would also provide an opportunity to label the many different geographic features and regions that the authors mention, and which readers in other parts of the world will not be familiar with.

Experimental design

The submission certainly describes original primary research within the scope of PeerJ. The research was conducted rigorously and ethically, and methods are described well.

The authors do need to clarify the methods very briefly in one place (explain how the MST results affect SKMC).

The introduction does not fully convey why the research question is meaningful and relevant, and the aims make the work seem less exciting than it is. The aims should be less ambiguous, and more specific about the different hypotheses about refugia and recolonization that are evaluated.

Validity of the findings

The data and analysis appear quite sound, and all the critical points on which the conclusions are based are included.

In one case, speculation is not clearly labeled, or a citation is missing, or the sentence is confusing, but I can't be sure which. (I have labeled this place in the pdf).

The conclusions are related to the aims, but the aims should ideally to be reworded to reflect the specific refugia hypotheses addressed.

Additional comments

This is nice work, with relevant findings, and I strongly recommend that it be accepted after some revisions to the writing.

The primary issue appears to be language, because the writing could benefit from more concise, focused, wording. (I have noted most of these places in the pdf).

Annotated reviews are not available for download in order to protect the identity of reviewers who chose to remain anonymous.

Reviewer 3 ·

Basic reporting

No Comments

Experimental design

I see no serious flaws in the analysis of the plastid data, I did not evaluated the methodological part of the niche distribution modelling. I have, however, several major concerns on the design and interpretation (particular the taxonomic implications) of the study. Please see the "General comments" section

Validity of the findings

The study brings new and interesting findings on evolutionary history of an important model species and I am convinced it will be finally highly suitable for wide audience in PeerJ.

Additional comments

1) There has been several phylographic studies on A. halleri from Pauwels et al. producing wealth of data on population-level diversity of plastid DNA haplotypes. It is very unclear to me why the authors did not included these data into their analyses(e.g. at least as a separate analysis using less markers, only those compatible across their studies, but a wider sampling to draw more robust conclusions), in particular when the authors of the previous studies are also included in the current team. For example, Pauwels et al 2012 shows a distinct group in most of the Southern Alps, which were completely omitted in this study. I see no serious problem if the current study is focused only on the thoroughly sampled Hercynian/W Carpathian populations, but such incomplete sampling hampers discussion e.g. on the refugia outside the sampled region.
2) Following the concerns raised in the point 1) and the general fact the study is based only on one, maternally-inherited, marker, I find quite speculative several conclusions on the evolutionary history of the group. In particular, (i) the existence of the Dinaric/Balkan refugium and ruling out the survival through last ice Age in the Carpathians is highly speculative when no data are present from most of the Romanian Carpathians and Southern Dinarids, where the species occur (see e.g. file S4) but was not sampled. Such speculations are in fact based only on the distributional modelling that is an useful indicative tool does not provide any firm evidence. (ii) The existence of a refugium north from the Alps is also speculative as the authors miss genetic data from most of the Eastern and southern Alps where the species is also abundant and existence of a foothill/Alpine refugium is still as likely as their proposed version. (iii) The minimum spanning tree also does not agree with the “simple” two-refugium scenario as the Eastern Carpathian chlorotypes are in the middle of the tree and the putative second refugium occupy, among others, three tip branches adjacent to the E Carpathian haplotypes (J,I,H). I thus highly recommend either broaden the sampling range by adding the already published data or considerably “boiling down” the conclusions on the history of the species outside the sampled areas. In particular the speculative conclusions on the absence of Carpathian refugium and existence of the Balkan one should be removed from the Abstract and the final conclusions and should remain only as a tentative HYPOTHESIS in the discussion.
3) The high diversity in the W Carpathians could not be explained based on recent gene flow only (see p 16 l. 19). The haplotype diversity of those populations is in generally high (highest among the compared regions, cf Table 5 !!), they contain a private haplotype F (see also p. 15 l. 20) and there are more diverse populations than only those containing the putatively “admixing” P chlorotype. I am thus not convinced the genetic data rules out the possibility of in situ glacial survival in W Carpathians (although it also does not prove it!). The postglacial recolonisation of the W Carpathians is thus suggested only by the niche modelling. Finally, it his highly confusing to me that in p. 21 l. 2 the authors finally go back to the W Carpathian populations and consider them highly distinct by following statement “The presence of this genetically quite distinct group may support the hypothesis on the existence of subsp. tatrica”. Please reconsider thus your conclusions on the survival of the Western Carpathian populations as well.
4) Finally, I see highly inappropriate to draw ANY taxonomic conclusions on plastidDNA-based dataset only. The maternally inherited marker brings only partial information on evolutionary history of a species, that is highly incomplete regarding processes such as hybridisation. In addition, a proper taxonomic revision should be based on a broader interdisciplinary re-evaluation of the study group (including morphology) and sampling throughout the range of the species. I thus find the current version as inappropriate and highly recommend removing completely the last section “Taxonomic implications” in order to avoid misleading interpretation from the Arabidopsis community. If the authors find it necessary to include such kind of discussion I see the dataset strong in questioning the Kolnik and Marholds 2006 concept in the Carpathians. I do not see, however, much explanatory power in drawing new taxonomic conclusions (such as defining the three groups on p.21 l. 15-23 and p 22) unless adding wealth of molecular and morphometric data.

Minor comments (page and line number indicated):
p2. l. 9-15 please re-formulate the abstract sections and remove the speculative conclusion of the southern Dinaric/Balkan refugia and recolonisation of the Eastern Carpathians as there are no genetic data presented on this in the current article.

p 4 l. 23 please cite the source on distributional data on A. halleri

p. 5 l. 2 information on distribution of the species in southern Dinarids is missing (cf map in file S4)

p. 6 l. 21. It would be highly suitable to publish such protocol, e.g. as a supplementary file, in order to aid repeatability of the method.

p. 9. l. 8-9 AMOVA should not be used to TEST the differentiation of a clustering method as it imposes a problem of circular evidence (see Meirmans 2015 Mol Ecol). I recommend presenting only the amounts of variation distributed within/among populations/groups in the particular partitions.

p 14. l. 1 It is unclear what the authors mean by “Balkan Mts.” Clearly not all mountains of the Balkan peninsula as the Dinaric Alps are put aside. please specify more clearly.

p 14. l. 20. It would be useful to add some other plant examples differentiating along the W-E Carpathian barrier that are not (sub) alpine but montane and such ecologically more similar too A. halleri, e.g. , Melampyrum sylavticum occurring in similar habitats (e.g. Tesitel et al. 2009 Preslia)

p. 16 l. 11 At least Delphinium oxysepalum is not endemic of the Tatra Mts only but also occurs in other sites in W Carpathians

p 22 l. 8-18 Conclusions highly duplicate with Abstract

Fig 2 Please specify the names of the discussed areas (Bohemian Forest, Western Carpathians etc.) into the map, it would help the reader unfamiliar with the geography of Central Europe.

---

## Round 0.2 · Minor Revisions

I sent the revision out to one prior reviewer who had a number of concerns; this reviewer feels you have adequately addressed those concerns, but notes (as I do, e.g. line 71 "within the investigated area", or line 200 "in order to assess the strength" or line 454-5 "make the hypothesis on the existence of a glacial refugium in this area" or line 504 "We have shown that the main genetic".) that there are a number of grammatical and typographical errors that remain in the manuscript.

I would also ask that you address a few additional minor revisions, originally bought up by the first round of review:

Please add geographic names to Figure 3.

I agree with not all the k values in figure 4 are necessary.

Please add a brief explanation of why MST and SKMC offer different signals is warranted.

Reviewer 3 ·

Basic reporting

The revised version of the article is sound and interesting. There are only minor typographic or gramatic errors throughout the text that should be checked prior the final publication, for example

l 308-309 - unclear acronym for Area under Receiver Operating Characteristic (ROC vs. AUC)
l 352 modelling
l 370 Pinus, Larix and Abies in italics

Experimental design

No Comments

Validity of the findings

No Comments

Additional comments

No Comments

---

## Round 0.3 · accepted · Accept

Please do a final round of checking spelling etc. I notice for example "asses" instead of "assess".

The reviewer very clearly understands the difference between haplotype network and clustering, but given the predominance of individual haplotypes within populations it would be surprising if the two gave very different results. In fact the results look pretty similar to me. I suspect the reviewer was wondering about the text which claims no ability to demarcate groups whereas the text claims the clustering clear separates populations in the E. Carpathians.